# Midbrain dopamine neurons sustain inhibitory transmission using plasma membrane uptake of GABA, not synthesis

Nicolas X Tritsch[1], Won-Jong Oh[2], Chenghua Gu[2], Bernardo L Sabatini[1]*

[1]Department of Neurobiology, Howard Hughes Medical Institute, Harvard Medical School, Boston, United States; [2]Department of Neurobiology, Harvard Medical School, Boston, United States

**Abstract** Synaptic transmission between midbrain dopamine neurons and target neurons in the striatum is essential for the selection and reinforcement of movements. Recent evidence indicates that nigrostriatal dopamine neurons inhibit striatal projection neurons by releasing a neurotransmitter that activates GABA$_A$ receptors. Here, we demonstrate that this phenomenon extends to mesolimbic afferents, and confirm that the released neurotransmitter is GABA. However, the GABA synthetic enzymes GAD65 and GAD67 are not detected in midbrain dopamine neurons. Instead, these cells express the membrane GABA transporters mGAT1 (Slc6a1) and mGAT4 (Slc6a11) and inhibition of these transporters prevents GABA co-release. These findings therefore indicate that GABA co-release is a general feature of midbrain dopaminergic neurons that relies on GABA uptake from the extracellular milieu as opposed to de novo synthesis. This atypical mechanism may confer dopaminergic neurons the flexibility to differentially control GABAergic transmission in a target-dependent manner across their extensive axonal arbors.

## Introduction

Dopamine (DA)-releasing neurons in the mammalian midbrain play an important role in fundamental behaviors, including motivation, reinforcement learning and motor control, and their dysfunction is associated with a wide range of neuropsychiatric disorders (*Wise, 2004*; *Costa, 2007*; *Schultz, 2007*; *Morris et al., 2009*; *Redgrave et al., 2010*). The major target of midbrain DA neurons is the striatum, a large subcortical structure implicated in the selection and reinforcement of motor actions. DA neurons located in the substantia nigra pars compacta (SNc) project mainly to the dorsal striatum (also known as the caudate and putamen, CPu), whereas those in the ventral tegmental area (VTA) innervate the ventral striatum (or nucleus accumbens, NAc), forming the nigrostriatal and mesolimbic pathways, respectively. The striatum controls motor behavior through two parallel output streams with opposing effects; the so-called 'direct' and 'indirect' pathways. Each pathway arises from distinct groups of GABAergic striatal projection neurons (SPNs) that differ, amongst other things, in their response to DA (*Gerfen and Surmeier, 2011*; *Tritsch and Sabatini, 2012*). By providing DA to the striatum, SNc and VTA neurons are believed to play a pivotal role in balancing the activity of direct- and indirect-pathway SPNs. However, our understanding of the cellular and molecular mechanisms employed by DA neurons to modulate striatal function remains incomplete.

Although the activation of metabotropic receptors following the release of DA from SNc/VTA neurons is undoubtedly central to their function, DA neurons also co-release several other transmitters that shape striatal output (*Hnasko et al., 2010*; *Stuber et al., 2010*; *Tecuapetla et al., 2010*; *Tritsch et al., 2012*). In particular, we recently showed that DA neurons in SNc potently inhibit action potential firing in SPNs by releasing a transmitter that activates GABA$_A$ receptors (*Tritsch et al., 2012*). Release

*For correspondence:
bsabatini@hms.harvard.edu

Competing interests: The authors declare that no competing interests exist.

**eLife digest** The electrical signals that are fired along neurons cannot be transmitted across the small gaps, called synapses that are found between neurons. Instead, the neuron sending the signal releases chemicals called neurotransmitters into the synapse. These neurotransmitters bind to receptor proteins on the surface of the second neuron and control how it fires.

A neurotransmitter called dopamine plays a key role in the circuits of the brain that control how we learn certain tasks involving movement. In particular, two populations of neurons from the midbrain that release dopamine target the striatum, an area of the brain that is responsible for motor control. These neurons also release other neurotransmitters, but the identity of these other chemicals is not known, and the details of the interaction between the neurons and the striatum are poorly understood.

Previous research showed that some of the midbrain neurons activate receptors that normally respond to a neurotransmitter called gamma-aminobutyric acid (GABA). However, several different chemicals can trigger this receptor. Using a range of techniques, Tritsch et al. now confirm that dopamine neurons release GABA alongside dopamine, and that this applies to both sets of the dopamine-producing neurons that feed into the striatum.

Some neurons can manufacture GABA from amino acids found in their internal fluid. However, Tritsch et al. could not detect the enzymes needed for this reaction in dopamine-producing neurons. Instead, these neurons contain proteins that can transport GABA across the cell membrane, which suggests that the neurons collect GABA from the extracellular fluid that surrounds them.

of this neurotransmitter requires activity of the vesicular monoamine transporter Slc18a2 (VMAT2), which can be replaced by exogenous expression of the vesicular GABA transporter Slc32a1 (also known as VGAT or VIAAT). In addition, VMAT2 expression in SPNs lacking VGAT is sufficient to sustain GABAergic transmission. Collectively, these findings suggest that SNc neurons co-release GABA using VMAT2 for vesicular loading. However, this study raises several important questions. First, does GABAergic transmission generalize to DA neurons in the VTA, or is it limited to nigrostriatal afferents? Second, what is the identity of the transmitter released by DA neurons? GABA exists as a zwitterion at neutral pH and lacks the characteristic molecular structure of VMAT2 substrates, which typically feature an aromatic ring and a positive charge (*Yelin and Schuldiner, 1995*). $GABA_A$ receptors can be activated by several naturally-occurring agonists and allosteric modulators, including β-alanine, taurine, imidazole-4-acetic acid and neurosteroids (*Johnston, 1996*; *Belelli and Lambert, 2005*). Although none of these molecules constitute ideal candidates for VMAT2-dependent vesicular transport based on their structure, charge, or both, they nonetheless raise the possibility that DA neurons liberate a transmitter other than GABA to inhibit SPNs. Third, do all DA neurons contribute to GABAergic signaling, or is it reserved to a subpopulation of cells, similar to glutamate co-release from DA neurons (*Hnasko and Edwards, 2011*). Finally, how is inhibitory synaptic transmission sustained in DA neurons? Addressing this point will help identify molecules required for GABAergic transmission by DA neurons and will permit the development of genetic tools to determine the relative contribution of SPN inhibition by DA neurons in vivo under normal and pathological conditions.

In this study, we address these questions by examining the cellular and molecular mechanisms that underlie the rapid $GABA_A$ receptor-mediated inhibition of SPNs upon stimulation of DA axons. Our results provide strong evidence that the inhibitory transmitter released by midbrain DA neurons is GABA, and suggest that GABA co-release is a general feature of all midbrain DA neurons. Moreover, we reveal that DA neurons rely on GABA uptake through the plasma membrane—but not on de novo GABA synthesis—to sustain GABAergic transmission.

## Results

### Stimulation of mesolimbic axons evokes inhibitory currents in striatal projection neurons

We recently reported that activation of SNc axons in dorsal striatum evokes monosynaptic, $GABA_A$ receptor-mediated inhibitory postsynaptic currents (IPSCs) in SPNs (*Tritsch et al., 2012*;

*Figure 1—figure supplement 1*). To determine whether this GABAergic signaling extends to DA neurons projecting to ventral striatum, we expressed channelrhodopsin 2 (ChR2) in the VTA using one of two approaches ('Materials and methods'). We either injected *Slc6a3-ires-Cre* mice (**Backman et al., 2006**) stereotaxically with an adeno-associated virus expressing Cre recombinase-dependent ChR2 (*Figure 1A*) or drove ChR2 expression genetically by crossing *Slc6a3-ires-Cre* mice to transgenic mice (**Madisen et al., 2012**) containing a conditional allele of ChR2 in the *Rosa26* locus (*Figure 1D*). Mice also harbored *Drd2-Egfp* or *Drd1a-tdTomato* bacterial artificial chromosome (BAC) transgenes to permit distinction between direct- and indirect-pathway SPNs, respectively (**Gong et al., 2003**; **Ade et al., 2011**). We performed whole-cell voltage-clamp recordings from SPNs in sagittal brain slices of NAc in the presence of inhibitors of ionotropic glutamate receptors (NBQX and R-CPP) and metabotropic GABA$_B$ receptors (CGP55845) to prevent excitatory synaptic transmission by dopaminergic axons, as well as modulatory effects of GABA$_B$ receptors, respectively. Under our recording conditions, optogenetic stimulation of VTA axons with brief (1 ms) flashes of blue light reliably evoked inward IPSCs in direct- and indirect-pathway SPNs that were blocked by the GABA$_A$ receptor antagonists picrotoxin (n = 12; *Figure 1B,E*) and SR95531 (n = 4, not shown). With the exception of synaptic latency, optogenetically-evoked IPSCs (oIPSCs) exhibited similar properties in both experimental systems and were consequently pooled for analysis. They averaged 353 ± 60 pA in peak amplitude (range = 68–905 pA; n = 16) and their kinetics were similar to those observed in dorsal striatum (*Tritsch et al., 2012*; *Figure 1—figure supplement 1*), with a 10–90% rise time of 2.4 ± 0.2 ms and a decay time constant of 46.8 ± 7.2 ms. The synaptic latency of oIPSCs in NAc did not differ from that

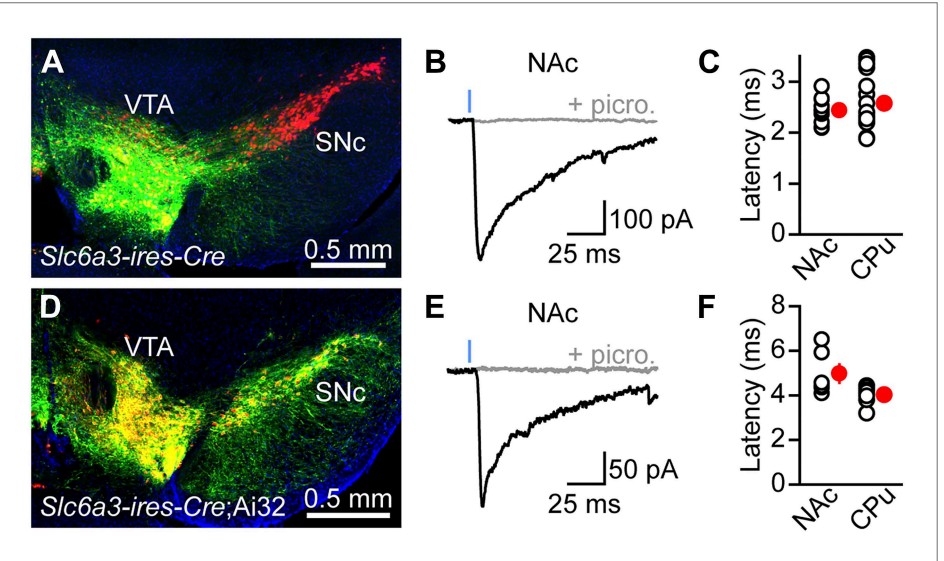

**Figure 1**. Stimulation of VTA axons evokes GABAergic currents in nucleus accumbens SPNs. ChR2 was expressed in VTA DA neurons virally (**A**–**C**) or genetically (**D**–**F**). (**A**) Coronal cross section of a *Slc6a3-ires-Cre* mouse ventral midbrain immunolabeled for TH (red) showing viral transduction of ChR2-EYFP in the VTA (green). (**B**) Representative oIPSC recorded from an indirect-pathway SPN in NAc before (black trace) and after (gray trace) bath application of the GABA$_A$ receptor antagonist picrotoxin (100 μM). Blue line depicts a 1-ms full field flash of 473 nm laser light (5 mW·mm$^{-2}$). All recordings in this and subsequent figures were performed at −70 mV using a high Cl$^-$ internal solution in the presence of NBQX (10 μM), R-CPP (10 μM), and CGP55845 (2–5 μM) in the perfusate. (**C**) Latency from flash onset to oIPSC onset in NAc and dorsal striatum (Caudate/Putamen, CPu) SPNs. For analysis of oIPSCs in CPu, ChR2 was expressed in SNc in a separate cohort of mice. White circles depict individual recordings, red circles are mean ± SEM. (**D**–**F**) As in (**A**–**C**) for recordings in *Slc6a3-ires-Cre*;Ai32 mice. Note that the synaptic latency in these mice is significantly longer than in virally transduced *Slc6a3-ires-Cre* mice in both CPu and NAc (p<0.001, Mann–Whitney test), presumably because of lower ChR2 expression in Ai32 mice.

The following figure supplements are available for figure 1:

**Figure supplement 1**. Properties of DA neuron oIPSCs in dorsal striatum.

observed in dorsal striatum under similar experimental conditions (*Figure 1C,F*), indicating that the connection is monosynaptic. Synaptic stimulation of midbrain DA neurons therefore directly engages GABA$_A$ receptors on SPNs in both dorsal and ventral striatum.

## Dopaminergic IPSCs are shaped by membrane GABA transporters

The chemical identity of a synaptic neurotransmitter is traditionally established by determining whether a synapse fulfills several necessary criteria. They include the presence of (1) presynaptic synthetic enzymes, (2) presynaptic vesicular transporters, (3) postsynaptic receptors, and (4) a biochemical mechanism for inactivation. In the case of GABAergic signaling at dopaminergic synapses, conditions 2 and 3 are respectively satisfied by the presence of VMAT2 in DA neurons and GABA$_A$ receptors in SPNs. For many central nervous system transmitters, condition 4 is mediated by diffusion and reuptake through plasma membrane transporters. Of the four genes identified in mice that encode high affinity membrane GABA transporters, *Slc6a1* is expressed predominantly throughout the brain (*Borden, 1996*; *Lein et al., 2007*). Its gene product, mGAT1, distributes mainly to presynaptic terminals of GABAergic neurons and serves to shape the amplitude and time course of IPSCs as well as regulate extrasynaptic levels of GABA (*Isaacson et al., 1993*; *Jensen et al., 2003*; *Overstreet and Westbrook, 2003*; *Conti et al., 2004*; *Chiu et al., 2005*; *Bragina et al., 2008*; *Kirmse et al., 2008*; *Cepeda et al., 2013*). Importantly, mGAT1 is highly selective for GABA and does not transport structurally similar GABA$_A$ receptor agonists such as β-alanine, taurine, and muscimol (*Johnston et al., 1978*; *Tamura et al., 1995*). Thus, we reasoned that if dopaminergic IPSCs are modulated by inhibition of mGAT1, it would provide compelling evidence that GABA is the neurotransmitter released.

To test this, we recorded dopaminergic oIPSCs from SPNs in dorsal striatum (*Figure 2*). Stimulation intensity was calibrated to evoke sub-maximal IPSCs (approximately 20% of IPSC peak amplitude at maximum intensity; mean: 232 ± 40 pA, n = 19) in order to minimize spillover of GABA at dopaminergic synapses (*Figure 2—figure supplement 1*). We compared the amplitude and kinetics of IPSCs recorded before (baseline) and during the first 3–4 min following bath application of control saline (ACSF; n = 9; *Figure 2A*) or saline containing SKF 89976A, a selective mGAT1 antagonist (n = 10; *Figure 2B*). In both conditions, the IPSC amplitude progressively declined to ~70% of baseline (p<0.01 each, Wilcoxon signed-rank test; *Figure 2C*), whereas the synaptic latency and 10–90% rise time remained unchanged relative to baseline (p>0.1, Wilcoxon signed-rank test). The decrease in amplitude is consistent with previous reports of activity-dependent rundown of GABA and DA release from midbrain DA neurons in vitro (*Schmitz et al., 2003*; *Tritsch et al., 2012*; *Ishikawa et al., 2013*) and did not differ between conditions (p=0.8, Mann–Whitney test; *Figure 2C*, *Figure 1—figure supplement 1G*), indicating that acute inhibition of mGAT1 does not interfere with neurotransmitter release or postsynaptic GABA$_A$ receptors. By contrast, mGAT1 antagonism significantly affected decay kinetics: whereas oIPSC decay time constants remained stable in control recordings (baseline: 33.6 ± 5.8 ms; 3–4 min after ACSF wash-in: 32.1 ± 5.8 ms; p=0.6, Wilcoxon signed-rank test), they were prolonged by a factor of 3 in the presence of SKF 89976A (baseline: 38.6 ± 6.0 ms; 3–4 min after SKF 89976A wash-in: 115.9 ± 26.3 ms; p=0.002, Wilcoxon signed-rank test; *Figure 2D*). These results indicate that mGAT1 activity normally shortens the duration of dopaminergic IPSCs, likely by clearing released neurotransmitter from the extracellular space.

In the striatum, mGAT1 controls ambient levels of GABA that evoke a tonic GABA$_A$ receptor-mediated conductance in SPNs (*Ade et al., 2008*; *Kirmse et al., 2008*; *Santhakumar et al., 2010*; *Cepeda et al., 2013*). Consistent with this, acute pharmacological inhibition of mGAT1 was accompanied by a significant increase in holding current caused by a threefold increase in tonic GABA current in both direct- and indirect-pathway SPNs (*Figure 2—figure supplement 2*). To exclude the possibility that the increase in oIPSC decay time constant stems from changes in recording conditions during bath application of SKF 89976A, we also monitored spontaneous (s) IPSCs as well as IPSCs evoked electrically within the striatum (eIPSCs; *Figure 3*). Unlike dopaminergic oIPSCs, neither the amplitude nor the kinetics of eIPSCs were significantly affected after acute mGAT1 inhibition (baseline amplitude: 304 ± 70 pA; 3–4 min after SKF 89976A wash-in: 339 ± 80 pA. Baseline 10–90% rise time: 1.3 ± 0.4 ms; 3–4 min after SKF 89976A wash-in: 1.3 ± 0.4 ms. Baseline decay time constant: 14.8 ± 2.6 ms; 3–4 min after SKF 89976A wash-in: 19.6 ± 4.9 ms; n = 9; p>0.1 for all, Wilcoxon signed-rank test; *Figure 3A,B*), despite a notable increase in holding current during the first few minutes following bath application of SKF 89976A (baseline: −82 ± 16 pA; SKF 89976A: −143 ± 21 pA; n = 9; p<0.001, Wilcoxon signed-rank test). Moreover, although the amplitude of sIPSCs decreased slightly

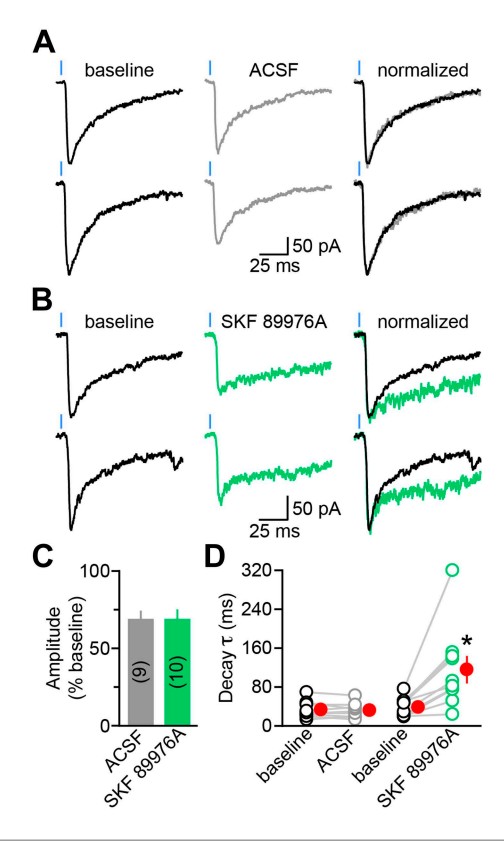

**Figure 2**. The decay kinetics of dopaminergic IPSCs are shaped by membrane GABA transporters. (**A**) Two representative oIPSCs recorded in SPNs using sub-maximal ChR2 stimulation (1ms; 0.3–2 mW·mm⁻²; blue line) before (baseline, left black trace) and 3–4 min after (middle gray trace) bath application of control saline (ACSF). *Right*, overlay of peak-normalized oIPSCs showing identical decay kinetics. (**B**) As in (**A**) for oIPSCs recorded in SKF 89976A (10 µM, in green). (**C**) Histogram of mean (±SEM) peak oIPSC amplitude normalized to baseline for SPNs perfused in ACSF (gray) and SKF 89976A (green). Number of recordings indicated in parentheses. (**D**) Plot of individual oIPSC decay time constants before and after bath application of ACSF and SKF 89976A. Mean (±SEM) shown in red. *p=0.002 vs baseline, Wilcoxon signed-rank test.

The following figure supplements are available for figure 2:

**Figure supplement 1**. oIPSC decay time constant increases with stimulus strength.

**Figure supplement 2**. mGAT1 controls ambient levels of GABA in the striatum.

in SKF 89976A relative to baseline (97 ± 9 vs 83 ± 7 pA, n = 14; p=0.01, Wilcoxon signed-rank test), the frequency and kinetics of sIPSCs remained unchanged (baseline frequency: 2.7 ± 0.5 Hz; 3–4 min after SKF 89976A wash-in: 2.5 ± 0.5 Hz. Baseline 10–90% rise time: 0.9 ± 0.1 ms; 3–4 min after SKF 89976A wash-in: 0.8 ± 0.1 ms. Baseline decay time constant: 6.8 ± 0.4 ms; 3–4 min after SKF 89976A wash-in: 6.4 ± 0.5 ms; p>0.05 for all, Wilcoxon signed-rank test; *Figure 3C,D*). These results are consistent with previous reports in the striatum and hippocampus (*Isaacson et al., 1993*; *Kirmse et al., 2008*). Thus, our results indicate that prolongation of light-evoked IPSCs by SKF 89976A is specific to dopaminergic synapses and not secondary to increased GABAergic tone. Together, these data reveal that the duration of dopaminergic IPSCs is critically dependent on mGAT1 function and, therefore, strongly support that GABA is the transmitter co-released by DA neurons.

## Mouse midbrain DA neurons do not express GABA synthetic enzymes

Previous reports indicate that up to 10% of midbrain DA neurons in rat contain detectable levels of mRNA for the 65 kDa isoform of glutamic acid decarboxylase (GAD65) (*Gonzalez-Hernandez et al., 2001*; *2004*). Together with GAD67, these enzymes constitute the major biosynthetic pathway for GABA in the central nervous system (*Soghomonian and Martin, 1998*). To determine the fraction of midbrain DA neurons capable of synthesizing (and by extension releasing) GABA in mice, we performed double fluorescence in situ hybridization for *Slc18a2* (*Vmat2*) and *Gad1* or *Gad2* (which encode GAD67 and GAD65, respectively). We focused our analyses to regions highlighted in *Figure 4A*, which consist of the SNc and lateral VTA. In agreement with previous findings (*Gonzalez-Hernandez et al., 2001*, *2004*), we did not detect significant overlap between *Vmat2* and *Gad1*, as only 4 out of 958 *Vmat2*+ neurons (0.4%) showed weak signal for *Gad1* in SNc and lateral VTA (*Figure 4A,B*). To our surprise, we were also unable to detect *Gad2* mRNA in these regions (*Figure 4D,E*): only 7 out of 1200 *Vmat2*+ cells (0.6%) were weakly *Gad2*+, despite the presence of numerous brightly-labeled GABAergic neurons in neighboring substantia nigra pars reticulata (SNr) and within the SNc and lateral VTA. We confirmed our ability to detect co-labeling by examining the histaminergic tuberomammillary nucleus (TMN) and dopaminergic A13 cell group (*Figure 4C,F*), both of which express GADs (*Lin et al., 1990*; *Esclapez et al., 1993*).

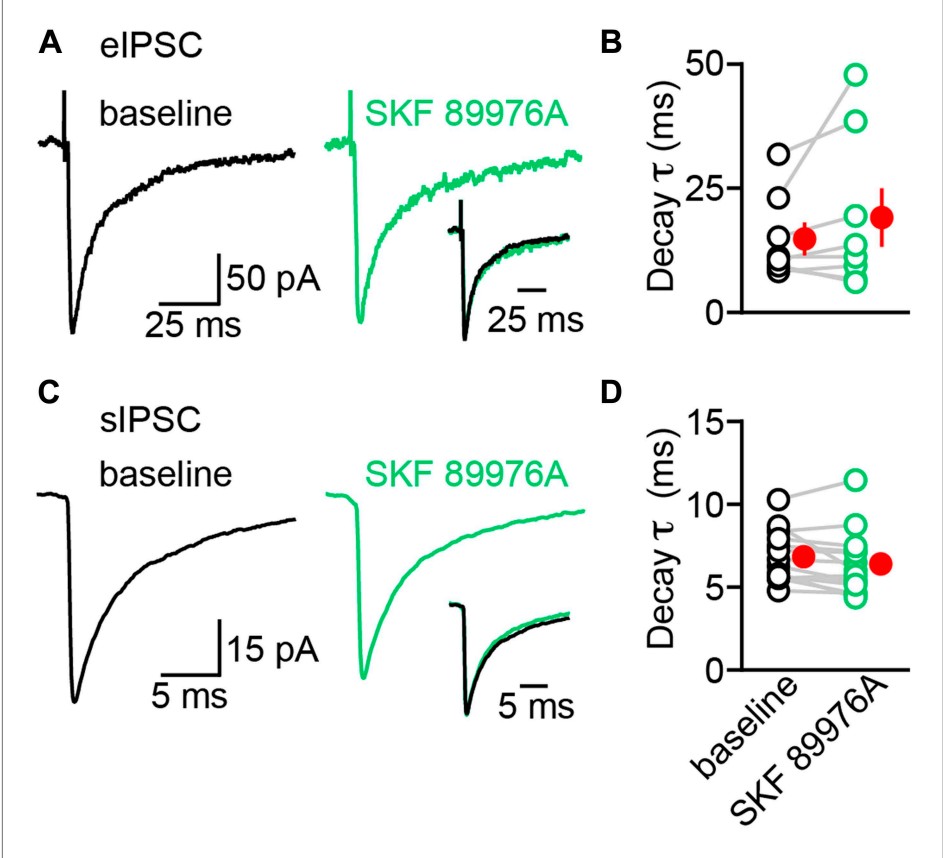

**Figure 3**. The decay kinetics of electrically-evoked and spontaneous IPSCs are insensitive to mGAT1 inhibition. (**A**) Representative electrically-evoked IPSC before (baseline, black trace) and 3–4 min after (green trace) bath application of SKF 89976A (10 μM). *Inset*, overlay of peak-normalized eIPSCs. (**B**) Plot of individual eIPSC decay time constants before (black) and after (green) SKF 89976A application. Mean (±SEM) shown in red. (**C** and **D**) As in (**A** and **B**) for spontaneous IPSCs. Note that sIPSCs have much faster kinetics compared to eIPSCs.

Levels of mRNA for GAD65 and GAD67 in DA neurons may be below the detection threshold, yet high enough to sustain GABA synthesis. To address this concern, we attempted to directly visualize GAD protein in DA neurons by immunofluorescence using antibodies directed against GAD65/67 and tyrosine hydroxylase (TH) to label catecholaminergic neurons. However, the high density of GABAergic axons converging into the substantia nigra, combined with the low concentration of GADs in cell bodies compared to axon terminals prevented us from clearly identifying DA neurons that expressed GADs (not shown). We instead imaged coronal brain sections from knock-in mice (*Tamamaki et al., 2003*; *Taniguchi et al., 2011*) expressing either EGFP or Cre recombinase under transcriptional control of the endogenous promoter for *Gad1* or *Gad2* (*Figure 5*). Cre expression was visualized using a sensitive fluorescent reporter allele (*Madisen et al., 2010*) and TH expression was revealed via immunofluorescence. These mice offer the advantage of having bright somatic labeling including, in the case of *Gad2-ires*-Cre mice, of cells with very little transcriptional activity. However, we were unable to detect any DA neuron expressing fluorescent reporter protein driven by either *Gad1* (0 out of 519 TH[+] neurons) or *Gad2* (0 out of 526 TH[+] neurons) promoters (*Figure 5A–D*). Collectively, these results indicate that midbrain DA neurons in the SNc and lateral VTA of mice do not express the GABA synthetic enzymes GAD65 and GAD67.

## Midbrain DA neurons express GABA transaminase

Mammalian cells possess alternative means of synthesizing GABA (*Seiler, 1980*; *Caron et al., 1987*; *Petroff, 2002*). GABA transaminase is an enzyme expressed at high levels in GABAergic neurons, including in the ventral midbrain that can reversibly convert succinate semialdehyde into GABA,

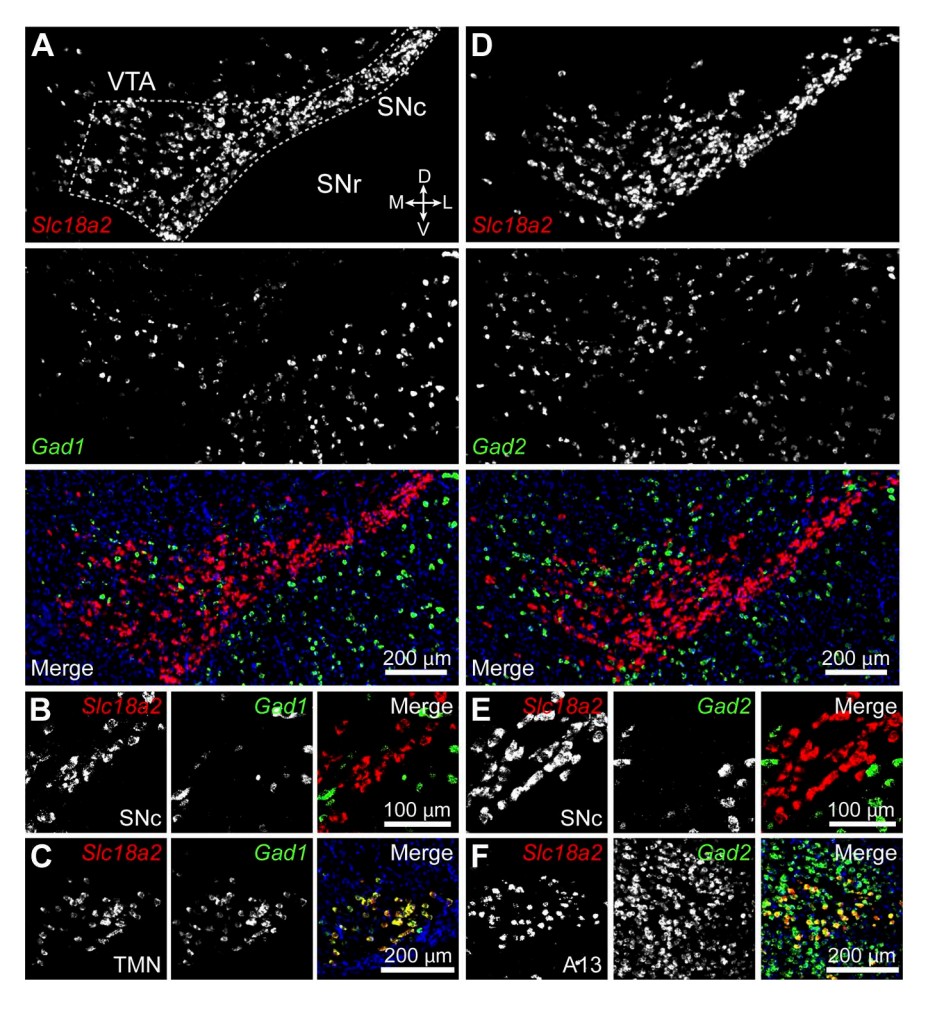

**Figure 4**. Midbrain DA neurons do not express *Gad1* or *Gad2*. (**A**) Two-color in situ hybridization of *Slc18a2* (*Vmat2*; *top*, red) and *Gad1* (*middle*, green) demonstrates the absence of co-labeled DA neurons (*bottom*) in a coronal section through lateral VTA and SNc (dashed outline). Nuclei are stained blue. SNr, substantia nigra pars reticulata; D, dorsal; V, ventral; M, medial; L, lateral. (**B**) Representative high magnification confocal image of *Slc18a2* (red) and *Gad1* (green) expression in SNc. (**C**) Double fluorescence in situ hybridization for *Slc18a2* and *Gad1* exhibits considerable overlap in the tuberomamillary nucleus (TMN). (**D** and **E**) As in (**A** and **B**) for *Slc18a2* (*Vmat2*) and *Gad2* expression. (**F**) *Slc18a2* and *Gad2* expression co-localize in the A13 dopaminergic cell group.

and vice versa (*Bessman et al., 1953*; *Roberts and Bregoff, 1953*; *Medina-Kauwe et al., 1994*). Because succinate semialdehyde is readily oxidized to succinate to sustain energy production as part of the Krebs cycle, GABA transaminase is thought to participate in the degradation of GABA in most cells, not its synthesis (*Balazs et al., 1970*). Nevertheless, we considered the possibility that GABA transaminase might contribute GABA in DA neurons. Double fluorescence in situ hybridization for *Vmat2* and *Abat* (the gene encoding GABA transaminase) revealed strong co-labeling in 701 out of 738 DA neurons (95%) within the SNc and lateral VTA (*Figure 6A,B*), indicating that DA neurons express genes involved in GABA metabolism. To test whether DA neurons employ GABA transaminase to synthesize GABA, we incubated slices of striatum in control ACSF or vigabatrin (VGT)—an irreversible inhibitor of GABA transaminase—for at least thirty minutes prior to recording oIPSCs in SPNs. In agreement with prior work (*Overstreet and Westbrook, 2001*), this manipulation significantly elevated extracellular GABA levels (picrotoxin-sensitive tonic current in SPNs in ACSF: 24 ± 3 pA, n = 11; in VGT: 115 ± 23 pA, n = 12; p<0.001, Dunn's Multiple Comparison Test), confirming the effectiveness of the drug. To minimize differences in light-evoked responses within experiments,

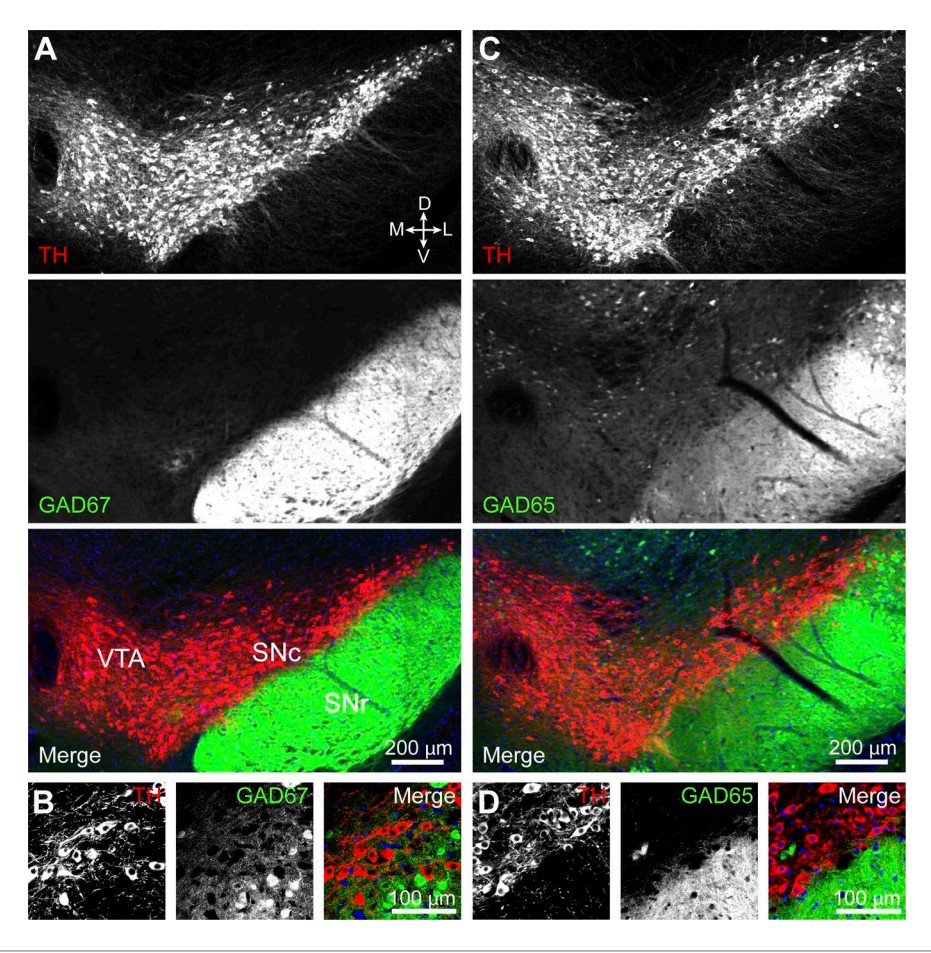

**Figure 5**. DA neurons in SNc/VTA do not express GAD65 or GAD67. (**A**) Low magnification epifluorescence image of a coronal section through the ventral midbrain showing the absence of overlap between tyrosine hydroxylase (TH) immunofluorescence (red) and endogenous EGFP (green) in the SNc and lateral VTA of *Gad1-Egfp* knock-in mice. Blue, nuclear stain. D, dorsal; V, ventral; M, medial; L, lateral. (**B**) Representative high magnification confocal image of SNc in *Gad1-Egfp* mice showing mutually exclusive expression of EFGP and TH, confirming that GAD67 is not expressed in SNc DA neurons. (**C** and **D**) As in (**A** and **B**) for coronal brain sections from *Gad2-ires-Cre* knock-in mice expressing a fluorescence Cre reporter allele (Ai14). TH immunolabeling and Cre reporter fluorescence respectively depicted in red and green for consistency. The absence of overlap indicates that GAD65 is not expressed in DA neurons of the SNc and lateral VTA.

we obtained maximal oIPSCs using strong ChR2 stimulation and limited comparisons to SPNs located in similar areas of dorsal striatum in adjacent slices. Under these conditions, we did not detect significant differences in oIPSC amplitude between both groups (ACSF: 1.1 ± 0.3 nA; VGT: 1.4 ± 0.2 nA; n = 14; p=0.3, Wilcoxon signed-rank test; *Figure 6C,D*), indicating that GABA transaminase function is not required for GABAergic signaling by DA neurons.

## Midbrain DA neurons express plasma membrane GABA transporters

Our data indicate that inhibitory synaptic transmission from DA neurons does not depend on synthesis of GABA by either GADs or GABA transaminase. In the central nervous system, several synapses rely on neurotransmitter reuptake across the plasma membrane as opposed to de novo synthesis to sustain synaptic transmission (*Torres et al., 2003*; *Edwards, 2007*). It is therefore conceivable that DA neurons inhibit SPNs by releasing GABA they acquire from the extracellular environment. mGAT1 and mGAT4 (encoded by *Slc6a11*) are the two major plasma membrane GABA transporters expressed in the mouse midbrain (*Borden, 1996*; *Lein et al., 2007*). We therefore performed double fluorescence

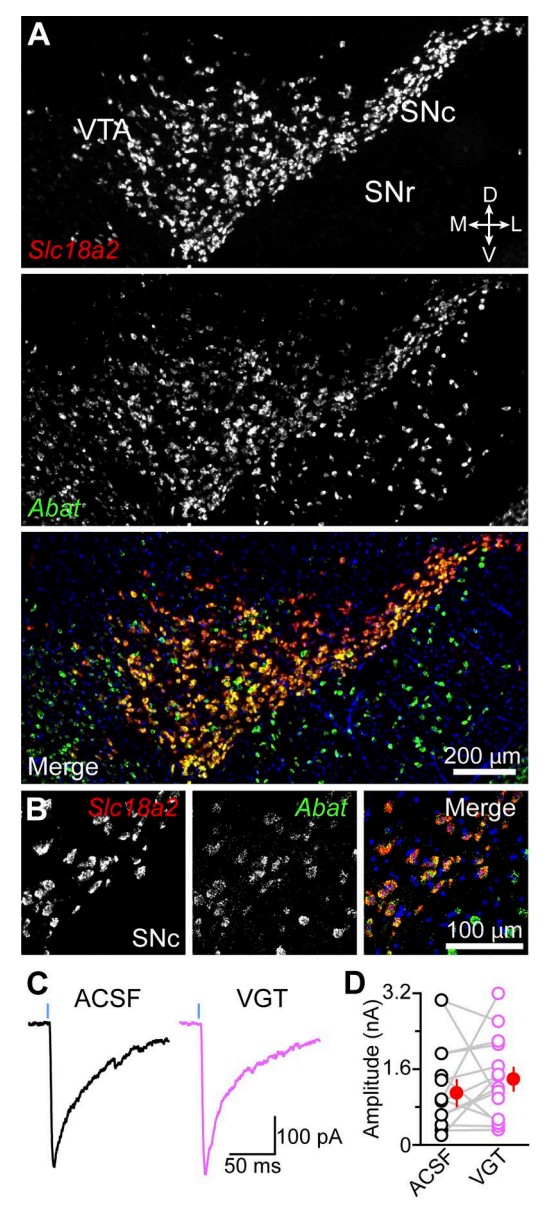

**Figure 6**. GABA transaminase is not required for GABA release by DA neurons. (**A**) Double fluorescence in situ hybridization for *Slc18a2* (*Vmat2*; red) and *Abat* (green) reveals that midbrain DA neurons overwhelmingly express GABA transaminase. Blue, nuclear stain. D, dorsal; V, ventral; M, medial; L, lateral. (**B**) High magnification confocal image of *Slc18a2* (red) and *Abat* (green) mRNA distribution in SNc. (**C**) Light-evoked IPSCs recorded from SPNs upon strong ChR2 stimulation (1 ms, 5 mW·mm⁻²; blue line) after prolonged incubation in ACSF (black) or vigabatrin (VGT, 100 µM; magenta). (**D**) Plot of mean peak oIPSC amplitude in SPNs recorded in similar regions of dorsal striatum in adjacent slices (depicted by gray lines) incubated in either ACSF (black) or VGT (magenta). Mean (±SEM) values for each group shown in red.

in situ hybridization for *Vmat2* and *mGat1* or *mGat4* to establish whether either isoform is present in DA neurons (*Figure 7*). Surprisingly, we found considerable expression of *mGat1* in 667 out of 748 *Vmat2*⁺ neurons (89%), particularly within the SNc. *mGat4* localized predominantly to glial cells, consistent with previous reports (*Borden, 1996*), although faint labeling above background was also detected in a substantial fraction (80%) of DA neurons in SNc and lateral VTA (568 out of 713 *Vmat2*⁺ neurons).

## GAT function is required for dopaminergic IPSCs

The expression of mRNA for mGAT1 and mGAT4 in DA neurons raises the possibility that DA neurons acquire GABA not through de novo synthesis, but rather through plasma membrane uptake. Being almost exclusively composed of GABAergic neurons, the striatum represents a rich source of extracellular GABA for dopaminergic terminals (*Ade et al., 2008*; *Kirmse et al., 2008*; *Janssen et al., 2009*; *Santhakumar et al., 2010*; *Cepeda et al., 2013*). Inhibition of mGAT1 for a few minutes does not affect the release of GABA from SNc axons (*Figure 2C*), indicating either that this pharmacological manipulation incompletely blocks GABA reuptake, or that it is too short to deplete presynaptic GABA levels. To determine whether uptake of ambient GABA is necessary for sustaining GABAergic transmission by DA neurons, we therefore inhibited both mGAT1 and mGAT4 in striatal slices using a cocktail of SKF 89976A and SNAP-5114 (a mGAT4 antagonist) for at least 30 min prior to examining oIPSCs in SPNs in the continued presence of GAT antagonists. Whereas oIPSCs obtained under control conditions averaged 1.3 ± 0.2 nA in amplitude (n = 15), oIPSCs recorded in adjacent slices with GABA transport chronically blocked were significantly smaller (0.20 ± 0.07 nA, n = 16; p<0.001 vs ACSF, Dunn's Multiple Comparison Test; *Figure 8A,B*). Importantly, prolonged exposure to SKF 89976A and SNAP-5114 did not prevent transmitter release non-specifically, as light-evoked dopamine release from SNc axons as well as GABA release from striatopallidal terminals were maintained under these conditions (*Figure 8—figure supplement 1*). Moreover, the small picrotoxin-sensitive oIPSCs that remained exhibited extremely long decay time constants

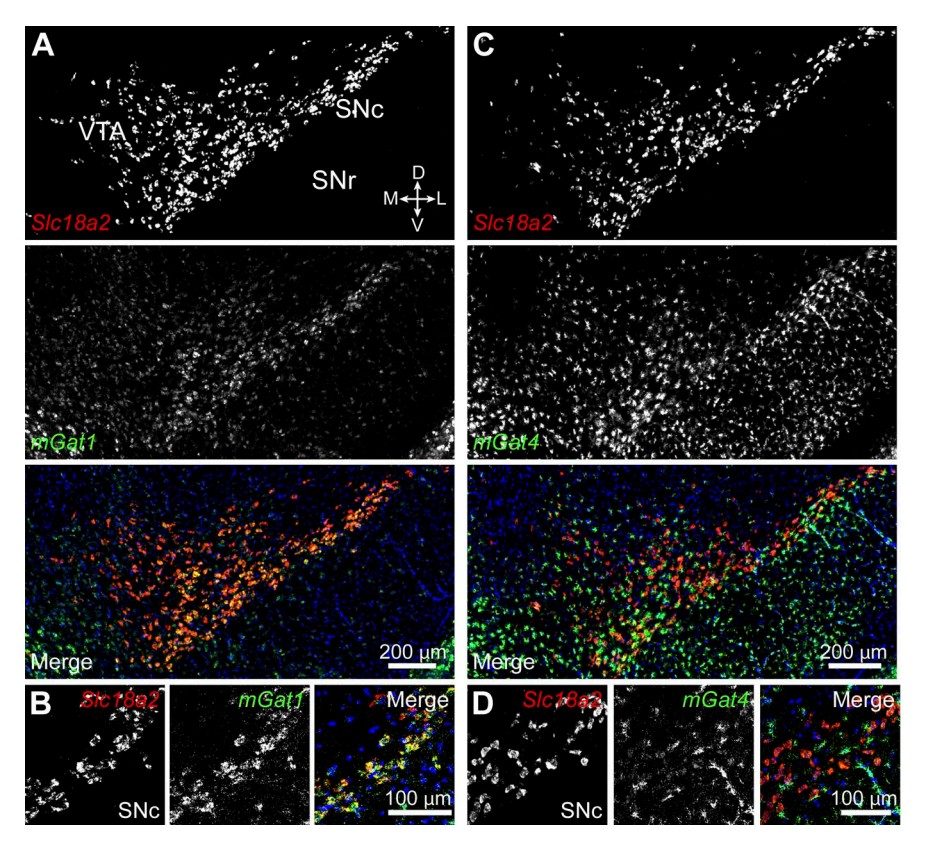

**Figure 7**. Midbrain DA neurons express plasma membrane GABA transporters. (**A**) Two color in situ hybridization for *Slc18a2* (*Vmat2*; *top*, red) and *mGat1* (*middle*, green) shows considerable overlap in SNc and lateral VTA (*bottom*). Nuclei are stained blue. D, dorsal; V, ventral; M, medial; L, lateral. (**B**) Representative high magnification confocal image of *Slc18a2* (red) and *mGat1* (green) in SNc confirms that DA neurons express mRNA for mGAT1. (**C**) Same as (**A**) for *Slc18a2* (*top*, red) and *mGat4* (*middle*, green). Note that mGat4 is most strongly expressed in star-shaped glial cells. (**D**) Confocal image through SNc reveals strong expression of *mGat4* mRNA in glial cells and weak labeling in DA neurons.

(>1.5 s), consistent with persistent GABAergic signaling in the absence of plasma membrane reuptake.

Chronic inhibition of GABA reuptake was also accompanied by considerable elevation of ambient GABA in the striatum, as evidenced by a 7.5-fold increase in tonic GABA$_A$ receptor-mediated current in SPNs (picrotoxin-evoked tonic current in ACSF: 24 ± 3 pA, n = 11; in GAT antagonists: 182 ± 32 pA, n = 11; p<0.001, Dunn's Multiple Comparison Test; *Figure 8C*). This increase in extracellular GABA might account for the reduced amplitude of oIPSCs by promoting the desensitization of GABA$_A$ receptors, by shunting synaptic currents, or both. To examine these possibilities, we included an additional experimental condition in which slices were pre-incubated in muscimol before evoking dopaminergic oIPSCs (*Figure 8A,B*). Muscimol is a potent synaptic and extrasynaptic GABA$_A$ receptor agonist that promotes GABA$_A$ receptor desensitization to the same extent as GABA (*Mortensen et al., 2010*), but displays minimal affinity for plasma membrane GABA transporters (*Johnston et al., 1978*). The concentration of muscimol was titrated to evoke tonic GABA$_A$ receptor-mediated currents similar those that developed upon persistent GAT inhibition (picrotoxin-sensitive tonic current: 212 ± 46 pA, n = 10; p<0.001 vs ACSF; p>0.05 vs GAT antagonists, Dunn's Multiple Comparison Test; *Figure 8C*). Strikingly, this manipulation did not reduce the amplitude of dopaminergic oIPSCs (1.3 ± 0.3 nA, n = 12; p>0.05 vs ACSF, p<0.01 vs GAT antagonists, Dunn's Multiple Comparison Test; *Figure 8A,B*), indicating that the reduction in oIPSC amplitude observed following GAT inhibition is not secondary to elevation of ambient GABAergic tone. In agreement with this conclusion, the amplitude of oIPSCs was not

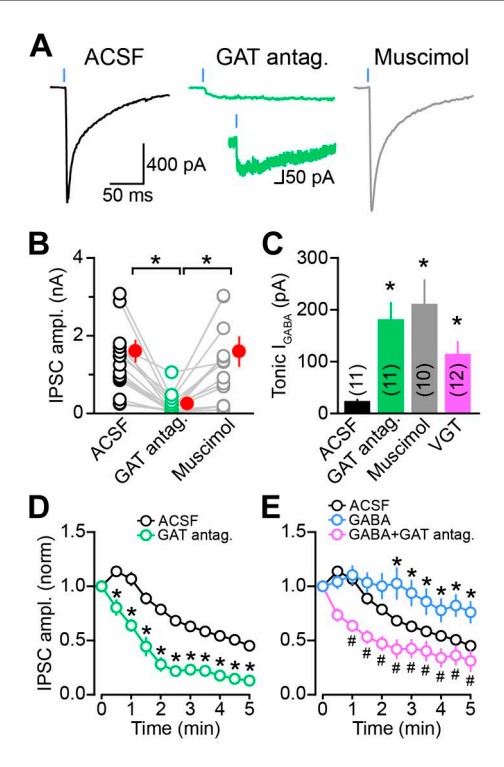

**Figure 8**. Sustained GABAergic signaling from DA neurons requires GAT function. (**A**) Dopaminergic IPSCs evoked by strong ChR2 stimulation (1 ms, 5 mW·mm⁻²; blue line) in slices incubated for at least 30 min in control ACSF (*left*, black), muscimol (0.1 µM; *right*, gray) or a cocktail of mGAT1 and mGAT4 antagonists (10 µM SKF 89976A + 50 µM SNAP-5114, respectively; *middle*, green). Recordings were performed in the continued presence of each drug, in addition to the GABA_B receptor antagonist CGP55845 (2–5 µM) and the glutamate receptor blockers NBQX (10 µM) and R-CPP (10 µM). *Inset*, oIPSC in GAT antagonists shown on longer time scale to illustrate slow kinetics. (**B**) Plot of peak oIPSC amplitudes recorded from individual SPNs in adjacent slices after prolonged incubation in ACSF (black), mGAT1 and mGAT4 inhibitors (green) and muscimol (gray). Mean (±SEM) indicated in red. *p<0.01 for indicated comparisons, Dunn's Multiple Comparison Test. (**C**) Histogram of tonic GABA current (I_GABA) measured in SPNs as the reduction in holding current evoked by bath application of the GABA_A receptor antagonist picrotoxin (100 µM) under control conditions (black), or after prolonged incubation in mGAT1 and mGAT4 blockers (10 µM SKF 89976A + 50 µM SNAP-5114; green), muscimol (0.1 µM; gray) or vigabatrin (100 µM; magenta). *p<0.01 vs ACSF, Dunn's Multiple Comparison Test. Number of recordings indicated in parentheses. (**D**) Plot of consecutive oIPSC amplitudes (normalized to the first light-evoked response) over time under control conditions (ACSF; black; n = 25) and after prolonged incubation in a cocktail of GAT antagonists
*Figure 8. Continued on next page*

diminished by prolonged treatment with VGT either (*Figure 6C,D*), despite significant increases in extracellular GABA (picrotoxin-sensitive tonic current in VGT: 115 ± 23 pA, n = 12; p<0.01 vs ACSF, p>0.05 vs GAT antagonists or muscimol, Dunn's Multiple Comparison Test; *Figure 8C*).

These results collectively indicate that GAT function is necessary for GABAergic transmission from SNc axons. If membrane GABA reuptake is required for supplying GABA for vesicular exocytosis, inhibiting GATs should prevent DA neurons from maintaining GABA release with repeated stimulation. Indeed, chronic inhibition of GATs precipitated the rundown of GABAergic transmission from DA neurons in an activity-dependent manner (*Figure 8D*, *Figure 8—figure supplement 2*). A second prediction is that elevating extracellular GABA should attenuate synaptic rundown. To test this, we locally applied GABA (100 µM) to the dorsal striatum for 5–10 min prior to recording oIPSCs from SPNs. Following this manipulation, the amplitude of oIPSCs was maintained for several minutes after break-in (*Figure 8E*). Importantly, this increase in oIPSC amplitude by exogenous GABA was prevented in slices incubated in GAT antagonists (*Figure 8E*). These data therefore provide strong evidence that membrane GABA transporters play an important role in sustaining GABAergic transmission from DA neurons by supplying the neurotransmitter GABA to presynaptic terminals for vesicular loading and exocytosis.

## Discussion

Despite the importance of midbrain DA neurons, the mechanisms by which they shape the activity of target neurons in the striatum remain poorly understood. In this study, we combined histochemical and electrophysiological approaches to define the cellular and molecular mechanisms that govern GABAergic signaling by DA neurons. Our findings indicate that the vast majority of midbrain DA neurons have the capacity to co-release GABA and identify presynaptic membrane GABA transporters as the likely mechanism employed by DA neurons to acquire GABA for vesicular exocytosis.

### Midbrain DA neurons co-release GABA

The chemical identity of a synaptic transmitter is typically established only after several independent observations converge on a plausible candidate. In this case, the transmitter released by midbrain DA neurons (1) functions as a GABA_A receptor agonist, (2) serves as a substrate for VGAT (*Tritsch et al., 2012*), and (3) is a substrate for mGAT1. Although VGAT can transport GABA, glycine, and

*Figure 8. Continued*

(10 µM SKF 89976A + 50 µM SNAP-5114; green; n = 11). *p<0.001 vs ACSF, Sidak's multiple comparison test. (**E**) As in (**D**) for slices supplied with exogenous GABA (100 µM) for 5–10 min before obtaining oIPSCs from SPNs in dorsal striatum (blue; n = 10), and slices supplied with exogenous GABA after prolonged inhibition of GATs with 10 µM SKF 89976A + 50 µM SNAP-5114 (magenta; n = 12). *p<0.05 vs ACSF, #p<0.005 vs GABA, Tukey's multiple comparison test. Control traces in (**D**) and (**E**) are the same as in *Figure 1—figure supplement 1D*.

The following figure supplements are available for figure 8:

**Figure supplement 1**. Chronic GAT block does not non-specifically affect synaptic transmission at GABAergic and dopaminergic synapses.

**Figure supplement 2**. oIPSC rundown is activity dependent.

β-alanine into synaptic vesicles (*Gasnier, 2000*; *Wojcik et al., 2006*; *Juge et al., 2013*), glycine does not function as a GABA$_A$ receptor agonist, and the low affinity of β-alanine for GABA$_A$ receptors would give rise to IPSCs with faster kinetics than the ones we observed (*Jones et al., 1998a*). In addition, mGAT1 is highly selective for GABA (*Borden, 1996*), and applying GABA exogenously helped sustain IPSCs from DA neurons in a GAT-dependent fashion. Although we did not detect GABA synthetic enzymes in DA neurons in our experiments, SNc and VTA neurons display a GABAergic phenotype, as they contain mRNA for GABA transaminase and the membrane GABA transporters mGAT1 and mGAT4. Together, these results strongly suggest that the neurotransmitter released by midbrain DA neurons is GABA. In agreement with this conclusion, a recent electron microscopic study detected GABA in close association with VMAT2-containing synaptic vesicles in dopaminergic terminals within striatum (*Stensrud et al., 2013*). Moreover, other dopaminergic neurons in the central nervous system have been reported to contain and/or release GABA, including amacrine cells in the retina (*Hirasawa et al., 2012*), periglomerular cells in the olfactory bulb (*Maher and Westbrook, 2008*; *Borisovska et al., 2013*; *Liu et al., 2013*), as well as dopaminergic cell groups throughout the brain (*Bjorklund and Dunnett, 2007*; *Stamatakis et al., 2013*), suggesting that GABA co-release may be a general property of dopaminergic cells.

## Midbrain DA neurons sustain inhibitory transmission using plasma membrane uptake of GABA

Synaptic transmission requires constant refilling of new and recycled vesicles, which is contingent upon the availability of transmitter in the cytosol. Most GABAergic neurons sustain inhibitory transmission by maintaining high cytosolic concentrations of GABA using de novo synthesis from glutamate by GAD67 and to a lesser extent GAD65 (*Asada et al., 1997*; *Tian et al., 1999*; *Petroff, 2002*). Given that all cells contain glutamate, an amino acid necessary for the production of proteins, the expression of either GAD67 or GAD65 in conjunction with a vesicular GABA transporter is sufficient to mediate vesicular release of GABA. Previous reports have estimated that up to 10% of DA neurons in the SNc and VTA of rats express mRNA for GAD65 (*Gonzalez-Hernandez et al., 2001, 2004*), suggesting that GABA release may be limited to a subpopulation of DA neurons. We were unable to replicate this observation using two separate approaches, indicating that DA neurons in the SNc and lateral VTA of mice do not rely on de novo synthesis of GABA and have instead adopted other means of obtaining GABA.

Most classical transmitters are transported back into the presynaptic terminal after vesicular release using plasma membrane transporters. This process is important not only for limiting the duration of synaptic currents and preventing extrasynaptic spillover, but also for replenishing presynaptic transmitter levels (*Edwards, 2007*; *Conti et al., 2011*). In fact, several synapses, including monoaminergic (*Giros et al., 1996*; *Bengel et al., 1998*; *Jones et al., 1998b*; *Xu et al., 2000*), glycinergic (*Gomeza et al., 2003a, 2003b*; *Rousseau et al., 2008*; *Apostolides and Trussell, 2013*) as well as some GABAergic terminals (*Mathews and Diamond, 2003*; *Bak et al., 2006*; *Fricke et al., 2007*; *Brown and Mathews, 2010*; *Wang et al., 2013*) rely heavily on membrane transporter function to maintain cytosolic transmitter pools available for vesicular loading. Our results show that DA neurons contain mRNA for mGAT1 and mGAT4, indicating that DA neurons might acquire GABA through membrane transport. Indeed, we find that prolonged inhibition of mGAT1 and mGAT4 inhibited GABAergic transmission by DA neurons and precipitated the rundown of oIPSCs, suggesting that DA neurons critically depend on extracellular GABA uptake to maintain cytoplasmic GABA levels and fill synaptic vesicles. By contrast, application of exogenous GABA helped sustain inhibitory transmission from DA neurons. By virtue of the fact that the vast majority of DA neurons in the SNc and lateral VTA contain mRNA for

mGATs, our results suggest that GABA co-release is, unlike glutamate co-transmission (*Hnasko and Edwards, 2011*), a common feature of these cells in the adult nervous system. The reliance on plasma membrane uptake may partially explain why release of both DA and GABA from these neurons is prone to rundown upon repeated stimulation in brain slices (*Schmitz et al., 2003*; *Tritsch et al., 2012*; *Ishikawa et al., 2013*), whereas release of glutamate from dopaminergic neurons or GABA from SPNs (which presumably depend on neurotransmitter synthesis to sustain release) do not suffer from the same shortcoming (*Tritsch et al., 2012*; *Figure 1—figure supplement 1D*). Nevertheless, these findings provide evidence that the expression of GABA synthetic enzymes is not required to sustain GABAergic transmission, and identify a GABAergic synapse that instead relies entirely on recycling extracellular GABA. Interestingly, a similar mechanism underlies exocytic release of serotonin from dopaminergic neurons (*Zhou et al., 2005*) as well as from glutamatergic thalamocortical neurons during development (*Lebrand et al., 1996*).

### Implications of GABAergic signaling for striatal function

Although DA neurons in SNc and VTA differ in the inputs they receive and the signals they convey (*Matsumoto and Hikosaka, 2009*; *Lammel et al., 2011*, *2012*; *Watabe-Uchida et al., 2012*; *Roeper, 2013*), we find that both cell populations are capable of co-releasing GABA with DA, pointing to a fundamental property of dopaminergic signaling. Morphological studies of SPNs at the electron microscopic level have revealed that the majority of dopaminergic synapses terminate on the neck of spines that receive excitatory inputs from cortex and thalamus (*Wickens and Arbuthnott, 2005*). Although the actions of DA are not believed to be spatially localized (*Arbuthnott and Wickens, 2007*), this unique synaptic arrangement is indicative of an additional, point-to-point mode of action. An intriguing possibility is that GABA co-release may function to hyperpolarize spines or shunt excitatory cortical and thalamic inputs by activating GABA$_A$ receptors. Phasic activation of DA neurons may thereby dampen ongoing cortical and thalamic drive onto SPNs to limit DA receptor-mediated plastic changes to synaptic inputs most strongly activated by salient or rewarding stimuli. In addition, the mechanism adopted by DA neurons to obtain GABA may confer these cells the flexibility to dynamically and locally control GABAergic transmission across their extensive axonal arbors. For instance, VTA neurons projecting to NAc and medial prefrontal cortex may only co-release GABA in the former, where extracellular GABA levels are high compared to cortex (*Drasbek and Jensen, 2006*; *Weitlauf and Woodward, 2008*). Alternatively, phasic and chronic changes in mGAT function or extracellular GABA levels resulting from synaptic activity, drugs or disease may alter the amplitude and kinetics of GABAergic currents arising from midbrain dopamine neurons. A greater understanding of the relative effects of DA and GABA on the activity of striatal circuits will help reveal how DA neurons contribute to behavior in health and disease.

## Materials and methods

### Experimental subjects and stereotaxic surgery

All experimental manipulations were performed in accordance with protocols approved by the Harvard Standing Committee on Animal Care following guidelines described in the US National Institutes of Health *Guide for the Care and Use of Laboratory Animals*. All mice were group-housed and maintained on a 12-hr light cycle with *ad libitum* access to food and water. *Slc6a3-ires-Cre* knock-in mice expressing Cre recombinase in DA neurons (*Backman et al., 2006*) were obtained from Jackson Labs (Bar Harbor, ME; stock # 006660). These mice were crossed with *Drd2-Egfp* (GENSAT, founder line S118) or *Drd1a-tdTomato* (stock # 016204; Jackson Labs) BAC transgenic mice to permit distinction between direct- and indirect-pathway SPNs (*Gong et al., 2003*; *Ade et al., 2011*). To genetically target indirect pathway SPNs, *Adora2a-Cre* BAC transgenic mice (GENSAT, founder line KG139) were used. For stereotaxic viral injections, postnatal day 18–25 mice were anesthetized with isoflurane, placed in a small animal stereotaxic frame (David Kopf Instruments, Tujunga, CA) and injected with 1 µl of an adeno-associated virus (~10$^{12}$ genome copies per ml; UNC Vector Core Facility, Chapel Hill, NC) encoding Cre-dependent ChR2 (AAV2/8.EF1α.DIO.hChR2(H134R)-mCherry or AAV2/8.EF1α.DIO.hChR2(H134R)-EYFP), as described previously (*Tritsch et al., 2012*). Injection coordinates were 0.8 mm anterior from Lambda, 1.3 mm lateral and 4.4 mm below pia for SNc, and 0.8 mm anterior from Lambda, 0.6 mm lateral and 4.4 mm below pia for VTA. Alternatively, ChR2(H134R)-EYFP was expressed genetically in Cre-containing cells by crossing *Slc6a3-ires-Cre;Drd2-Egfp* or *Slc6a3-ires-Cre;Drd1a-tdTomato* mice

with Ai32 mice (stock # 012569; Jackson Labs) (*Madisen et al., 2012*). The knock-in lines used in *Figure 5* include *Gad1-Egfp* (*Tamamaki et al., 2003*) and *Gad2-ires-Cre* mice (*Taniguchi et al., 2011*). The latter were crossed with mice bearing a Cre-dependent tdTomato reporter transgene (Ai14; stock # 007914; Jackson Labs) to reveal the distribution of Cre$^+$ cells (*Madisen et al., 2010*). Mice were maintained on a C57BL/6 background. Only mice heterozygous for all transgenes were used for experiments.

## Immunocytochemistry and in situ hybridization

Mice were anesthetized with isoflurane and transcardially perfused with phosphate buffered saline followed by 4% (wt/vol) paraformaldehyde in 0.1 M sodium phosphate buffer. 50-micrometer coronal brain sections were subject to immunohistochemical staining for tyrosine hydroxylase (AB152; 1:2000; Millipore, Billerica, MA), as described previously (*Tritsch et al., 2012*). Endogenous tdTomato and EGFP fluorescence were not immuno-enhanced. Double fluorescence in situ hybridization was performed using a tyramide signal amplification method according to the manufacturer's instructions (NEL753001KT; PerkinElmer, Waltham, MA), as previously described (*Kwon et al., 2012*). Briefly, brains from 4-week old mice were dissected and immediately frozen in liquid nitrogen. They were then cut in 25-µm-thick sections with a cryostat (Leica, Buffalo Grove, IL), postfixed in 4% PFA, acetylated in 1% triethanolamine and 0.25% acetic anhydride, dehydrated serially in ethanol, bleached in 3% hydrogen peroxide diluted in methanol, prehybridized, and hybridized at 65°C using the following anti-sense probes provided by the Allen Institute for Brain Science (riboprobe ID specified in parentheses): *Gad1* (RP_040324_01_F01), *Gad2* (RP_071018_02_B07), *Vmat2* (RP_071218_02_B08), mouse *Gat1* (RP_071204_04_H01), mouse *Gat4* (RP_050428_04_D04), and *Abat* (RP_050301_02_C02). For in vitro transcription, each cDNA was cloned out from a mouse brain cDNA library using the primer sets suggested in the in situ hybridization data portal from the Allen Institute for Brain Science. Two fluorescein- or digoxigenin-labeled riboprobes generated by an in vitro transcription method (Promega, Madison, WI) were hybridized simultaneously and stained by fluorescein or Cy3 chromogens, respectively. After staining, sections were mounted with Prolong Gold antifade reagent with DAPI nuclear stain (Life Technologies, Grand Island, NY).

## Slice electrophysiology

Acute brain slices and whole-cell voltage-clamp recordings from identified SPNs were obtained using standard methods, as described previously (*Tritsch et al., 2012*). Briefly, mice (49–178 days old; median = 107 days) were anesthetized and perfused with ice-cold artificial cerebrospinal fluid (ACSF) containing (in mM) 125 NaCl, 2.5 KCl, 25 NaHCO$_3$, 2 CaCl$_2$, 1 MgCl$_2$, 1.25 NaH$_2$PO$_4$ and 11 glucose (295 mOsm·kg$^{-1}$). Parasagittal slices of striatum (275-µm thick) were subsequently obtained in cold choline-based cutting solution (in mM: 110 choline chloride, 25 NaHCO$_3$, 2.5 KCl, 7 MgCl$_2$, 0.5 CaCl$_2$, 1.25 NaH$_2$PO$_4$, 25 glucose, 11.6 ascorbic acid, and 3.1 pyruvic acid). Following 15 min recovery in ACSF at 34°C, slices were kept at room temperature (20–22°C) until use. All solutions were constantly bubbled with 95% O$_2$/5% CO$_2$. Whole-cell voltage-clamp recordings were established from direct- and indirect-pathway SPNs in dorsal or ventral striatum resting 18–75 µm below the slice surface (median: 40 µm) in ACSF warmed to 32–34°C. Direct-pathway SPNs were identified as tdTomato$^+$ cells in *Drd1a-tdTomato* mice or EGFP$^-$cells in *Drd2-EGFP* mice, and indirect-pathway SPNs as tdTomato$^-$or EGFP$^+$ cells in *Drd1a-tdTomato* and *Drd2-EGFP* mice, respectively. Patch pipettes (2–4 MΩ) were filled with (in mM) 125 CsCl, 10 TEA-Cl, 10 HEPES, 0.1 Cs-EGTA, 3.3 QX-314 (Cl$^-$salt), 4 Mg-ATP, 0.3 Na-GTP, 8 Na$_2$-Phosphocreatine (pH 7.3 adjusted with CsOH; 295 mOsm·kg$^{-1}$). The recording perfusate always contained NBQX (10 µM) and R-CPP (10 µM) to block AMPA and NMDA receptor-mediated inward currents, respectively, as well as CGP55845 (2–5 µM) to prevent GABA$_B$ receptor-evoked pre- and post-synaptic modulation. For some experiments (*Figures 6 and 8*, *Figure 8—figure supplements 1 and 2*), slices were incubated for at least 30 min (and up to two hours) in ACSF containing CGP55845 (2–5 µM) in addition to vigabatrin (100 µM), muscimol (0.1 µM), or SKF 89976A (10 µM) + SNAP-5114 (50 µM) to allow for the depletion of cytosolic GABA levels and synaptic vesicles containing GABA, and to control for the effects of chronically elevated GABAergic signaling. In these cases, drugs continued to be present in the recording chamber for the duration of the experiment. For *Figure 8E*, GABA (100 µM, in ACSF containing either 5 µM CGP55845 or 5 µM CGP55845 + 10 µM SKF 89976A + 50 µM SNAP-5114) was locally applied to the slice via a gravity-fed 250-µm-wide flow pipe for 5–10 min prior to recording. For all voltage-clamp experiments, errors due to the voltage drop across the series resistance (<20 MΩ) were left uncompensated. Membrane potentials were corrected for a ~5-mV liquid junction potential. Under these conditions, GABA$_A$ receptor-mediated currents appeared

inward when SPNs were held at negative membrane potentials ($V_{hold}$ = −70 mV). To activate ChR2-expressing fibers, light from a 473-nm laser (Optoengine, Midvale, UT) was focused on the back aperture of the microscope objective to produce wide-field illumination of the recorded cell. Brief pulses of light (1-ms duration; 5 mW·mm$^{-2}$ under the objective for maximal stimulation, 0.3–2 mW·mm$^{-2}$, for sub-maximal stimulation) were delivered at the recording site at 30 s intervals under control of the acquisition software. Epifluorescence illumination was used sparingly to minimize ChR2 activation prior to recording. eIPSCs were evoked using constant-current pulses (0.1 ms, 7–90 µA) delivered every 15 s through a bipolar tungsten stimulating electrodes positioned within striatum, 100–200 µm away from the recorded cell. Constant-potential amperometry was performed as before (*Tritsch et al., 2012*) using commercial glass-encased carbon-fiber microelectrodes (Carbostar-1, Kation Scientific, Minneapolis, MN) placed within dorsal striatum slices and held at 600 mV. All recordings were obtained within 4 hr of slicing. All pharmacological agents were obtained from Tocris (Minneapolis, MN).

## Data acquisition and analysis

Brain sections processed for in situ hybridization or immunofluorescence were imaged with an Olympus VS110 slide-scanning microscope. High-resolution images of regions of interest were subsequently acquired with a Zeiss LSM 510 META confocal microscope (Harvard NeuroDiscovery Center). Individual imaging planes were overlaid and quantified for colocalization in ImageJ (NIH) and thresholded for display in Photoshop (Adobe). Confocal images in figures represent maximum intensity projections of 3-µm confocal stacks. Membrane currents were amplified and low-pass filtered at 3 kHz using a Multiclamp 700B amplifier (Molecular Devices, Sunnyvale, CA), digitized at 10 kHz and acquired using National Instruments acquisition boards and a custom version of ScanImage (*Pologruto et al., 2003*; available upon request or from https://openwiki.janelia.org/wiki/display/ephus/ScanImage) written in MATLAB (Mathworks). Electrophysiology data were analyzed offline using Igor Pro (Wavemetrics, Natick, MA). In figures, voltage-clamp traces represent the averaged waveform of 3–5 consecutive acquisitions. Averaged waveforms were used to obtain current latency, peak amplitude, 10–90% rise time and decay time constant. The latter was estimated by measuring the time elapsed from the peak of the IPSC to 1/$e$ (36.8%) of the peak amplitude. Detection threshold for sIPSCs was set at 40 pA to facilitate event detection in the presence of large and noisy tonic GABA currents. For pharmacological analyses in *Figures 2 and 3*, the peak amplitude of IPSCs measured 3–4 min following the onset of drug perfusion were averaged, normalized to baseline averages obtained immediately prior to drug application and compared statistically to values obtained at corresponding times in control preparations bathed in ACSF. Data (reported in text and figures as mean ± SEM) were compared using Prism 6 (GraphPad, La Jolla, CA) with the following non-parametric statistical tests (as indicated in the text): Mann–Whitney for group comparisons, Wilcoxon signed-rank for paired comparisons, and Kruskal–Wallis analysis of variance (ANOVA) followed by Dunn's Multiple Comparison Test for multiple group comparisons. In experiments characterizing the time course of synaptic transmission rundown, two-way ANOVAs were used followed by the Sidak's and Tukey's multiple comparison tests for comparisons between two and more conditions, respectively. p values smaller than 0.05 were considered statistically significant. N values represent the number of recorded cells. For most experiments, a single cell was recorded from in each slice, with each animal contributing fewer than five recordings to individual data sets.

## Acknowledgements

We thank R Shah, N Mulder, and R Pemberton for technical support and members of the Sabatini laboratory for discussions throughout the course of this study.

## Additional information

### Funding

| Funder | Grant reference number | Author |
| --- | --- | --- |
| National Institutes of Health | NS046579 | Bernardo L Sabatini |
| Nancy Lurie Marks Family Foundation | | Nicolas X Tritsch |
| Lefler Postdoctoral Fellowship | | Won-Jong Oh |

| Funder | Grant reference number | Author |
|---|---|---|
| Alice and Joseph Brooks Fund | | Won-Jong Oh |
| Howard Hughes Medical Institute | | Bernardo L Sabatini |
| National Institutes of Health | NS064583 | Chenghua Gu |

The funders had no role in study design, data collection and interpretation, or the decision to submit the work for publication.

## Author contributions
NXT, BLS, Conception and design, Acquisition of data, Analysis and interpretation of data, Drafting or revising the article; W-JO, Acquisition of data, Analysis and interpretation of data; CG, Conception and design, Drafting or revising the article, Contributed unpublished essential data or reagents

## Ethics
Animal experimentation: This study was performed in strict accordance with the recommendations in the Guide for the Care and Use of Laboratory Animals of the National Institutes of Health. All experimental manipulations were performed in accordance with protocols approved by the Harvard Medical Area Standing Committee on Animal Care (#03551).

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
