## [Decision Letter]

Thank you for sending your work entitled “Midbrain dopamine neurons sustain inhibitory transmission using plasma membrane uptake of GABA, not synthesis” for consideration at *eLife*. Your article has been favorably evaluated by a Senior editor and 3 reviewers, one of whom is a member of our Board of Reviewing Editors. The Reviewing editor and the other reviewers discussed their comments before reaching this decision, and the Reviewing editor has assembled the following comments to help you prepare a revised submission.

This relatively straightforward report follows up on prior work of these authors examining the corelease of GABA from “dopamine” neurons. In this report they extend their studies from the nigrostriatal pathway to the mesolimbic pathway. Interestingly GABA is supplied not by synthetic enzymes in the neurons, but rather by expression of membrane GABA transporters. As previously shown, vesicular uptake of GABA can be performed by the vesicular monamine transporter VMAT2. The questions addressed in this manuscript are whether such corelease occurs in the mesolimbic pathway and whether the agonist is GABA or a related analog.

The authors nicely demonstrate in Figure 1 that VTA inputs to NAc have similar properties with respect to GABA corelease as the previously demonstrated SNc inputs to dorsal striatum. The majority of the figures then focus on the identity and source of GABA in the VTA-NAc pathway. These experiments are carefully conducted and well presented and convincingly demonstrate that GABA is the principal agonist and that the main source is membrane uptake rather than intracellular synthesis. Although these results are somewhat predictable from prior studies and could be further extended, the results are important in understanding the rather novel, albeit still somewhat mysterious, role of a fast-acting transmitter in midbrain dopamine neurons, and thus should be of interest to a broad community of neuroscientists. The substantial comments of the 3 reviewers are numbered below.

1) These data make several specific, testable predictions that would enhance the strength of the main interpretation. For example, increasing extracellular GABA should increase the amplitude of DA-IPSCs through an mGAT-dependent mechanism. Does bath application of GABA, or perhaps extracellular GABA uncaging, increase DA-IPSCs? And is this effect blocked by the acute presence of mGAT blockers during said GABA application? [87] (cited) showed that GAT activity can rescue transmission in conventional GABAergic synapses when GABA is bath applied in the presence of gabazine. This experiment would reveal the timecourse over which DA neurons take up GABA from the extracellular space and package it into synaptic vesicles, and it would be important to contrast the timecourse of the authors' novel mechanism with that of enzymatic synthesis at more traditional GABA synapses. This experiment would enhance also the claim that GAT supplies the GABA used for synaptic vesicles.

2) A second prediction is that rundown of transmission induced by SKF should be activity dependent. I found it confusing at first that Figure 2 shows little effect of SKF on IPSC amplitude and Figure 7 shows a large effect, and I am sure readers will also be caught off guard. The difference is that with time the synapses in SKF are depleted of GABA. Therefore if the synapses were stimulated one might expect to accelerate the process, confirming that it reflects depletion of vesicular GABA rather than a side effect of the drug.

3) The authors deal with a number of potential confounding factors such as the increased tonic GABA currents produced by GABA transporter inhibition, showing that this does not affect the kinetics of quantal GABA events from sources other than dopamine neurons. On the other hand, it is remarkable that the inhibitor does not affect the kinetics of these other events – is it so selective for GABA release from dopamine neurons because the transporter is only expressed on those terminals? It would be helpful if the authors could address this surprising result – previous work has indicated expression of GAT-1 by SPNs and an effect of a different inhibitor (NO-711) on amplitude and decay time of events evoked by local depolarization (47).

4) Treatment with GAT-1 inhibitor for longer times effectively eliminates the GABA release by dopamine neurons, as anticipated if it depends on the GABA taken up from other cells. However, it is also possible that dopamine neurons make GABA themselves, and the main argument used here against this is the apparent lack of expression by a variety of criteria. Although these seem reasonable, other work cited by the authors in a different context (77) along with other work they do cite suggests expression of GAD by a subset of dopamine neurons. Rather than rely on the anatomy, it would help greatly to demonstrate that inactivation of GAD specifically in dopamine neurons does not eliminate the dopaminergic IPSCs. This might require a substantial amount of work that would lie outside the scope of this manuscript (for example a double knockout of both isoforms that is conditional and may not even be available), but it would certainly be helpful. However, showing that expression of a GAD isoform in dopamine neurons makes the dopaminergic IPSCs resistant to inhibition of GAT-1 would be considerably simpler.

5) The authors also suggest that GABA corelease involves both mesostriatal and mesolimbic projections, in contrast to the apparent corelease of glutamate by a subset of VTA neurons. However, [77] also showed that only ∼10 % of striatal terminals apparently contain GABA, raising the question whether the dopaminergic IPSCs derive from a subset of both populations, or all of them. It would be great if the authors could address this point, and one way might be to determine whether GAT-1 inhibition influences dopamine release measured by voltammetry-GABA corelease by many terminals should have some effect on total dopamine release, but involvement of only a subset would produce minimal if any effect. This would also help to address one important functional outcome of corelease that has not yet been examined.

---

## [Author Response]

We are thankful for the reviewers’ prompt evaluation of our manuscript and for their valuable comments and suggestions. We are delighted that our work was reviewed favorably and described as “straightforward”, “carefully conducted” and “well-presented”. We agree with the reviewers that the results described in our study are an important and necessary first step to elucidate the mysterious role of fast-acting neurotransmission from midbrain dopamine neurons. However, we take exception with the notion that the principal findings of this study were in any way predictable from prior work. Although our initial study (85) pointed to GABA as being the transmitter released by dopamine neurons, we acknowledged that other molecules might perform similar functions, and remained agnostic as to the identity of the co-released transmitter as we initiated and carried out the present study. This was particularly true when we failed to detect GABA synthetic enzymes in midbrain dopamine neurons. Moreover, we are not aware of any prior evidence that might have predicted that midbrain dopamine neurons acquire GABA through membrane uptake as opposed to intracellular synthesis.

Please find below our response to each of the reviewer’s comments. We hope the reviewers will find our revised manuscript improved and suitable for publication.

*1) These data make several specific, testable predictions that would enhance the strength of the main interpretation. For example, increasing extracellular GABA should increase the amplitude of DA-IPSCs through an mGAT-dependent mechanism. Does bath application of GABA, or perhaps extracellular GABA uncaging, increase DA-IPSCs? And is this effect blocked by the acute presence of mGAT blockers during said GABA application?*
[87]
*(cited) showed that GAT activity can rescue transmission in conventional GABAergic synapses when GABA is bath applied in the presence of gabazine. This experiment would reveal the timecourse over which DA neurons take up GABA from the extracellular space and package it into synaptic vesicles, and it would be important to contrast the timecourse of the authors' novel mechanism with that of enzymatic synthesis at more traditional GABA synapses. This experiment would enhance also the claim that GAT supplies the GABA used for synaptic vesicles*.

We performed the experiment suggested by the reviewers, in which we show that bath application of GABA increases the amplitude of IPSCs evoked by dopaminergic neuron stimulation in a GAT-dependent manner. These data are consistent with the claim that GABA is the transmitter released, and that it is supplied by GATs for vesicular release. These results are now shown in Figure 8, and discussed in the text.

Technical limitations unfortunately prevented us from performing the second series of experiments, which would have allowed us to contrast the time course of GABA uptake and vesicular filling in dopamine neurons with that of ‘classical’ GABAergic synapses that depend on GABA synthesis. The studies by [87] were performed in culture and the authors benefited from stable IPSC amplitudes, as well as the ability to rapidly apply and remove extracellular solutions containing GABA, gabazine and SKF89976A. In our experiments in which, the axons are, by necessity, severed from the cell bodies, the amplitude of dopaminergic IPSCs runs down considerably with time (see discussion below and Figure 1—figure supplement 1) and complete washout of gabazine is slow (tens of minutes) and variable in duration, effectively preventing us from carrying out similar analyses. It is also unclear that we would be able to assign differences in time course to enzymatic synthesis vs. membrane uptake, as classical GABAergic synapses and this one also rely on distinct vesicular transporters (VGAT and VMAT2, respectively), which most certainly differ in transport kinetics.

*2) A second prediction is that rundown of transmission induced by SKF should be activity dependent. I found it confusing at first that*
Figure 2
*shows little effect of SKF on IPSC amplitude and*
Figure 7
*shows a large effect, and I am sure readers will also be caught off guard. The difference is that with time the synapses in SKF are depleted of GABA. Therefore if the synapses were stimulated one might expect to accelerate the process, confirming that it reflects depletion of vesicular GABA rather than a side effect of the drug*.

We apologize for the confusion and thank the reviewers for pointing out the need to clarify the description of our experimental manipulations. As the reviewers indicate, the major difference between Figures 2 and 7 is the duration of drug application. Figure 2 describes the acute effects of SKF89976A within the first few minutes of bath application. Under these conditions, we did not observe any effect of SKF89976A (or SKF89976A + SNAP-5114; See Figure 1—figure supplement 1) on the amplitude of oIPSCs. This indicates that blocking membrane GABA reuptake does not immediately interfere with the ability of dopaminergic terminals to release GABA, presumably because synaptic vesicles are already filled and cytoplasmic GABA levels have not yet been depleted. In Figure 8, slices were incubated for thirty or more minutes in SKF89976A + SNAP-5114 to allow for depletion of cytoplasmic (and by extension vesicular) GABA in dopaminergic synapses. We have modified the text to stress this distinction.

Following the reviewers’ suggestion, we now provide evidence in Figure 8 that the rundown process is accelerated following chronic GAT blockade, and in Figure 8—figure supplement 1 and Figure 8—figure supplement 2 that the effect of this manipulation on oIPSC amplitude does not reflect a side effect of the drug. Specifically, we show that, in addition to reducing the amplitude of optogenetically-evoked IPSCs, prolonged GAT blockade significantly accelerates rundown of GABAergic transmission. We did not detect differences in either the amplitude of IPSCs or the time course of rundown between slices that were incubated in SKF89976A + SNAP-5114 for less than one hour or between one and two hours, indicating that rundown is activity dependent. Importantly, preventing membrane GABA uptake for thirty minutes or longer does not prevent synaptic release of dopamine from nigrostriatal axons or GABA from striatopallidal terminals indicating that SKF89976A and SNAP-5114 do not impair synaptic transmission non-specifically. These results are therefore consistent with the hypothesis that chronic block of membrane GABA reuptake selectively inhibit GABAergic transmission at dopaminergic synapses by depleting cytosolic GABA levels.

*3) The authors deal with a number of potential confounding factors such as the increased tonic GABA currents produced by GABA transporter inhibition, showing that this does not affect the kinetics of quantal GABA events from sources other than dopamine neurons. On the other hand, it is remarkable that the inhibitor does not affect the kinetics of these other events – is it so selective for GABA release from dopamine neurons because the transporter is only expressed on those terminals? It would be helpful if the authors could address this surprising result – previous work has indicated expression of GAT-1 by SPNs and an effect of a different inhibitor (NO-711) on amplitude and decay time of events evoked by local depolarization (*[47]*)*.

The different sensitivity of IPSCs originating from dopamine neurons vs. ‘classical’ GABAergic neurons to SKF89976A is indeed remarkable, pointing to fundamental structural and/or biochemical differences between both synapses. It is unlikely that selective expression of mGAT1 in dopamine neurons fully accounts for this difference, as mGAT1 is expressed widely throughout the brain, including in the striatum. However, it is possible that dopaminergic synapses rely exclusively on mGAT1 for membrane GABA transport, whereas classical GABAergic synapses may utilize other transporters in the presence of SKF89976A, such as mGAT4. Indeed, the tonic GABA current evoked by acute bath application of SKF89976A and SNAP-5114 (a mGAT4 antagonist) is more than twice that of SKF89976A alone (picrotoxin-evoked current in SKF89976A: 90 ± 9 pA, n = 15 SPNs; in SKF89976A+SNAP-5114: 198 ± 29 pA, n = 10 SPNs; P = 0.005, Mann Whitney test). Additional factors may contribute to differences in the sensitivity of IPSC kinetics to GAT blockers, including GABAA receptor subunit composition (1), GABAA receptor desensitization (Overstreet et al. 2000; [63]), stimulation strength (38) and synapse density (64), amongst others.

The reviewers correctly note that another group (47) previously reported changes in the amplitude and kinetics of electrically-evoked IPSCs in striatum following mGAT1 inhibition. In that study, the observed decrease in IPSC amplitude was attributed to presynaptic GABAB receptors, which were pharmacologically blocked in our experiments. The effect on IPSC kinetics consisted of a slight increase (13 %) in the time taken for the current to decay to 10 % of the peak amplitude in P12-14 slices recorded at ambient temperature. It is unclear whether a significant effect would have been detected using more classical measures of decay time, or if this effect would be maintained in older animals and at elevated temperatures. Importantly, GABAergic IPSCs are particularly heterogeneous in their sensitivity to GAT blockade, which is strongly influenced by experimental conditions (38; 64).

*4) Treatment with GAT-1 inhibitor for longer times effectively eliminates the GABA release by dopamine neurons, as anticipated if it depends on the GABA taken up from other cells. However, it is also possible that dopamine neurons make GABA themselves, and the main argument used here against this is the apparent lack of expression by a variety of criteria. Although these seem reasonable, other work cited by the authors in a different context (*[77]*) along with other work they do cite suggests expression of GAD by a subset of dopamine neurons. Rather than rely on the anatomy, it would help greatly to demonstrate that inactivation of GAD specifically in dopamine neurons does not eliminate the dopaminergic IPSCs. This might require a substantial amount of work that would lie outside the scope of this manuscript (for example a double knockout of both isoforms that is conditional and may not even be available), but it would certainly be helpful. However, showing that expression of a GAD isoform in dopamine neurons makes the dopaminergic IPSCs resistant to inhibition of GAT-1 would be considerably simpler*.

[77] used immunogold EM to localize GABA within TH^+^ and VMAT2^+^ terminals in the striatum of mice and rats, but did not examine the distribution of GADs. We only know of two studies having directly investigated whether SNc and VTA dopamine neurons express GADs (33, 34). These studies reported that while 6-10 % of midbrain dopamine neurons in rat contain detectable levels of mRNA for GAD65, their soma is immuno- negative for GABA and GAD65, indicating that SNc/VTA neurons may not be able to synthetize GABA. Our in situ hybridization data in wildtype mice, in combination with our immuno-histochemical analyses in transgenic reporter animals are consistent with this conclusion, as both approaches convincingly showed that dopamine neurons in the SNc and lateral VTA do not express the GABA synthetic enzymes GAD65 or GAD67.

In the revised manuscript, we provide additional functional evidence for GATs being necessary for GABAergic transmission in dopaminergic neurons, and include control experiments showing that chronic inhibition of GATs does not non-specifically impair vesicular release from midbrain dopamine neurons, or GABA release from a classical GABAergic synapse. Thus, we are currently unaware of any compelling anatomical or functional evidence in favor of midbrain dopamine neurons directly synthetizing GABA for release using GADs.

Global and conditional knockout mice for GAD65 and GAD67 exist, respectively (Asada et al*.* 1996; Obata et al*.* 2008), but we currently do not have access to them. Carrying out the suggested GAD knockout experiment would require a substantial amount of work (breeding mice harboring 5 transgenic alleles: Gad1^f/f^; Gad2^-/-^; DatIRES-Cre/wt) that is not motivated by prior or current evidence for GAD expression in midbrain dopamine neurons. We therefore agree with the reviewers that this experiment is beyond the scope of this manuscript.

Although the suggested gain-of-function experiment is considerably simpler, it too requires experimental tools (viral vectors encoding GADs) that are currently not at our disposal and would take several months to generate and characterize. We agree that it would nicely demonstrate that the inhibition of oIPSCs by GAT antagonists does not stem from unintended effects of these drugs on synaptic vesicle exocytosis or GABA synthesis by GADs. However, this experiment would, by itself, offer little additional insight as to the molecular mechanism normally responsible for providing GABA to midbrain dopamine neurons. Control experiments are now provided in Figure 8—figure supplement 1, which demonstrate that blockade of GATs has no effect on dopamine release from SNc axons and no effect on GABA release from striatopallidal terminals. In light of these findings, we hope that the reviewers agree that the results of the proposed gain-of-function experiment would not affect our conclusions, and that this experiment is therefore not essential to substantiate the claim that GATs are necessary for sustaining GABAergic transmission in midbrain dopamine neurons.

*5) The authors also suggest that GABA corelease involves both mesostriatal and mesolimbic projections, in contrast to the apparent corelease of glutamate by a subset of VTA neurons. However,*
[77]
*also showed that only ∼10 % of striatal terminals apparently contain GABA, raising the question whether the dopaminergic IPSCs derive from a subset of both populations, or all of them. It would be great if the authors could address this point, and one way might be to determine whether GAT-1 inhibition influences dopamine release measured by voltammetry-GABA corelease by many terminals should have some effect on total dopamine release, but involvement of only a subset would produce minimal if any effect. This would also help to address one important functional outcome of corelease that has not yet been examined*.

The study by [77] is remarkable in that it convincingly reveals the existence dopaminergic terminals in the striatum of mice and rats that contain a high density of immunogold particles for GABA. However, immunogold EM is a challenging technique that severely underlabels molecules of interest, and therefore does not easily lend itself to absolute measures of expression: the fact that 10 % of dopaminergic terminals contained enough immunogold particles to qualify as being GABAergic cannot be interpreted to mean that 90 % do not contain GABA. Instead, their data indicate 1) that dopaminergic terminals contain the neurotransmitter GABA, 2) that GABA is present in or near VMAT^+^ synaptic vesicles, and 3) that the concentration of GABA is large enough to be detected in a sizeable fraction of dopaminergic terminals.

Nevertheless, the question raised by the reviewers is important, as establishing whether GABA co-release is common to all midbrain dopamine neurons or a unique feature of a subset of cells is essential to determine its functional relevance. It should be noted that regardless of the molecular mechanisms engaged or the fraction of midbrain dopamine neurons implicated, GABA co-transmission may be a general feature of monoaminergic neurons, as opposed to an exception, suggesting an important role for dopamine-GABA co-transmission (see references in this manuscript and [85]).

Determining whether optogenetically-evoked GABAergic currents arise from release in all terminals, or from a fraction of terminals is challenging, and it is unclear how the suggested electrochemical experiment would address this point. We currently do not have evidence pointing to a GABAergic modulation of dopamine exocytosis (measured by amperometry): light-evoked dopamine release is unaffected by inhibition of GABAA receptors (85) or by activation of GABAB receptors with baclofen (5 μM; unpublished observation). Moreover, we now provide amperometry data showing that dopamine release is not changed after chronic inhibition of GAT (Figure 8—figure supplement 1), indicating that neither buildup of extracellular GABA in the slice, nor depletion of cytosolic GABA in dopaminergic terminals significantly affect dopamine release.

The suggestion that GABA co-release in the striatum may apply to all midbrain dopamine neurons (as opposed to a subset of SNc and VTA neurons) is based on several observations. First, oIPSCs are large and present in every SPN studied to date, as opposed to glutamatergic events, which are significantly smaller in amplitude and only observed in a fraction SPNs (85). Second, minimum stimulation experiments suggest that oIPSCs in SPNs represent compound responses to numerous small-amplitude GABAergic inputs as opposed to a few large ones (85). Third, mGat1 mRNA was detected in the vast majority of midbrain dopamine neurons, suggesting that most cells have the ability to take up GABA from the extracellular space in GABA-rich brain regions such as the striatum. Although not definitive, these experiments strongly suggest that GABA co-release is a general feature of most dopaminergic terminals in the striatum.